# Development of an Intelligent Tablet Press Machine for the In-Line Detection of Defective Tablets Using Machine Learning and Deep Learning Models

**DOI:** 10.3390/pharmaceutics17040406

**Published:** 2025-03-24

**Authors:** Sun Ho Kim, Su Hyeon Han

**Affiliations:** 1College of Pharmacy, Dankook University, 119, Dandae-ro, Dongnam-gu, Cheonan-si 31116, Republic of Korea; 2Department of Mechanical Engineering, Kongju National University, 1223-24, Cheonan-daero, Seobuk-gu, Cheonan-si 31080, Republic of Korea; tngus8581@gmail.com

**Keywords:** intelligent tablet press machine, defective tablet sorting, machine learning, deep learning, process analytical technology

## Abstract

**Objectives:** This study aims to develop a tablet press machine (TPM) integrated with machine learning (ML) and deep learning (DL) models for in-line detection of defective tablets as a Process Analytical Technology (PAT) tool. This study aimed to predict tablet defects, including capping occurrence and inappropriate tablet breaking force (TBF), using real-time processing data. **Methods:** Free-flowing metformin HCl (MF) granules produced using the granulation method were compressed into tablets using a TPM. Commercial-scale experiments were conducted to determine the MF tablets’ defect criteria. Random Forest (RF) and Artificial Neural Network (ANN) models were designed and trained using sensed in-line data, including compression force, ejection force, and compression speed, to predict tablet quality defects. Subsequently, the TPM was designed and manufactured for in-line PAT using an RF model. The TPM was verified by sorting defective tablets in-line using a pretrained defect-detection algorithm. **Results:** The RF model demonstrated the highest predictive accuracy at 93.7% with an Area Under the Curve (AUC) of 0.895, while the ANN model achieved an accuracy of 92.6% with an AUC of 0.878. The TPM successfully sorted defective tablets in real time, achieving 99.43% sorting accuracy and a defective tablet detection accuracy of 93.71%. **Conclusions:** These results suggest that a ML-based TPM applied during the tableting process can detect defects non-destructively during the scale-up of wet granulation. In particular, it can serve as the base TPM model for an in-line PAT process during a scale-up process that produces small batches of multiple products, thereby reducing additional labor, time, and API consumption, and decreasing environmental pollution.

## 1. Introduction

As the pharmaceutical industry is highly regulated by regulatory institutions such as the FDA, final products must meet stringent specifications, which require advanced quality control systems in the pharmaceutical industry [1]. In addition, the pharmaceutical industry is currently under significant pressure to reduce costs and minimize the quantity of raw materials required, which requires more efficient and scientific approaches [2]. However, traditional pharmaceutical manufacturing is typically conducted using batch processing, which is a time-consuming form of offline laboratory testing conducted on post-sampling to evaluate quality control (QC), which can only be verified at the end of the process, leading to the final products being at risk of being discarded, resulting in energy consumption and environmental pollution [3]. Furthermore, controlling the input variables that could prevent defects during the scale-up process is challenging because of simultaneous changes in many parameters, such as compression force, compression speed, tablet machine type, ejection pressure, and ejection speed [4].

To meet the requirements of pharmaceutical regulations, pharmaceutical companies have recently developed systematic and scientific methods in smart factories to predict product quality through non-destructive, in-line inspections with minimal effort [5]. Recent advancements in analytical techniques, terahertz pulsed imaging (TPI), and image analysis have enabled the in-line and online monitoring of pharmaceutical coating processes, facilitating the in-line control of critical parameters such as film thickness and uniformity [6]. In solid formulations in pharmaceutical processes, near-infrared (NIR) spectroscopy has been evaluated as a process analytical technology (PAT) tool for the in-line monitoring of freeze-drying processes, revealing complementary roles in detecting critical process parameters such as ice nucleation, mannitol crystallization, and ice sublimation [7]. De Beer et al. evaluated Raman spectroscopy as a PAT tool for in-line end-point monitoring during powder blending processes, demonstrating its ability to assess blend homogeneity and enhance process understanding when combined with experimental design [8]. Furthermore, Raman spectroscopy was combined with machine learning (ML) techniques for monoclonal antibody (mAb) production and developed as a PAT tool for the in-line monitoring of charge variants, achieving accurate quantification of species and total protein concentrations [9]. Although existing PAT technologies such as TPI, Raman spectroscopy, and NIR spectroscopy are generally non-destructive and capable of in-line monitoring, the existing systems not only require expensive additional equipment but also rely on online analysis for parameters such as moisture content and component analysis. However, these technologies find it difficult to measure in-line tableting parameters, such as compression force, during the actual tableting process in pharmaceutical manufacturing. Additionally, these systems have limitations such as complex data interpretation, particularly when analyzing multiple variables simultaneously and handling complex datasets. Therefore, this study developed a ML-based TPM using simple and cost-effective load sensors. The ML-based TPM offers significant enhancements in handling complex, multivariate data and improving predictive accuracy.

Most AI-based in-line PAT studies within the pharmaceutical field have been conducted using ML techniques based on a software-based impractical theoretical method. To the best of our knowledge, a deep learning (DL) study for defect detection in the tableting process has not yet been reported, due to limited experimental data. In a previous study, prediction models for critical quality attributes (CQA) related to tablet compression were evaluated using ML [10], but due to the limitations of commercial tablet press machines (TPM), the AI model could not be applied to in-line TPM. Accordingly, in this study, artificial neural network (ANN) and random forest (RF) models were further assessed using advanced techniques, including DL, and a TPM was designed and fabricated to detect defective tablets, specifically inappropriate tablet breaking force (TBF) and capping defects. Additionally, verification was conducted to support the practical hardware-based implementation of this TPM. For preparing tablets, metformin HCl (MF) treatment for type 2 diabetes was used [11]. The compression force, ejection force, and tablet thickness were sensed in-line during the tableting process in the TPM. The models were trained using ANN and RF, and model performance was evaluated. Subsequently, the TPM was validated by implementing an actual tablet process, and its performance in sorting defective tablets using a programmed model algorithm was assessed.

## 2. Materials and Methods

### 2.1. Materials

MF (median diameter: 38.33 μm, purity >99 *w*/*w*%) was procured from Granules India Limited (Madhapur, Hyderabad, India), as previously described in [10]. Other excipients, including polyvinylpyrrolidone (PVP K30), methacrylic acid copolymers (Eudragit S100), high-viscosity-grade HPMC2208, Carbomer 934P (Carbopol^®^ 934PNF), and Magnesium Stearate (MgSt), were obtained from BASF (Ludwigshafen Land, Rheinland-Pfalz, Germany), Evonik (Essen, NRW, Germany), Dow Chemicals (Montgomeryville, PA, USA), BF-Goodrich (Cleveland, OH, USA), and FACI Asia Pacific (Merlimau Pl., Jurong Island, Singapore), respectively. Ferric oxide red was purchased from Univar (Billericay, Essex, UK).

### 2.2. Tablet Manufacturing and Testing Method

#### 2.2.1. Morphological and Physical Characteristics of MF Granules

The MF-loaded granules were formulated with an approximate drug content of 80%, and their physical characteristics were evaluated as previously described [12]. Granules included carbomer 934P, HPMC2208, and methacrylate copolymers for sustained release to reduce rapid plasma clearance following oral administration [13]. Particle size was measured using a particle size analyzer with a median diameter of 38.3 μm (Table 1), and the granules were appropriately desiccated, with a loss on drying (LOD) of 0.74% *w*/*w*. The Hausner ratio (HR) and Compressibility Index (CI) values of the granules were determined to indicate “excellent” or “good” levels of flowability. These free-flowing granules were then used to manufacture MF-loaded tablets to evaluate the TPM’s ability to detect defects using ML and DL-based models.

#### 2.2.2. Preparation of MF-Loaded Granules via Wet Granulation and Tablet Compression

The MF-loaded granules were manufactured via the wet granulation method to prepare a batch of 300,000 tablets, as described in [14]. The specific composition of the mixture used in the granulation process was prepared with reference to a previous study [10] and is listed in Table 2. MF-loaded granules were prepared by dissolving binders in a water–ethanol solution, which was then sprayed onto the drug powder, followed by blending with excipients, drying, and sieving. MgSt, comprising 0.6–2.0% of the total mixture, was sieved through a 40-mesh sieve, added to the granules, and lubricated at 10 rpm for 5 min.

#### 2.2.3. Research Procedure

The procedure for the design of defect prediction models using RF and ANN and their application to the TPM for in-line defect detection, as part of PAT, was conducted as shown in Figure 1. First, raw materials were formulated and manufactured using the wet granulation method (Figure 1a). Subsequently, tablets were prepared under various conditions to determine defective criteria while in-line recording of tableting parameters, including compression force, ejection force, and compression speed, was carried out. Subsequently, TBF, friability, and thickness were measured to analyze their correlation with the tableting parameters (Figure 1b). The collected data were then analyzed using RF and ANN algorithms with training and validation to develop an in-line defect-detection model (Figure 1c). A TPM was newly designed and manufactured (Figure 1d), and an RF model was installed on a computer embedded within the machine (Figure 1e). The TPM was then used for defect sorting and verification (Figure 1f).

#### 2.2.4. Preparation of MF-Loaded Tablets

The granules were loaded into a die and compressed to prepare the tablets using a TPM (TDP5, LFA, Taichung, Taiwan), as described previously [10]. The TPM was equipped with a commercial 9.8 mm-diameter round-type convex die and Euro-standard D441 punch. Tablet ejection and compression forces were measured at speeds between 80 and 107 mm/s using a 10,000 lb-capacity load cell (LCM375, FUTEK, Irvine, CA, USA) with a minimum resolution of 0.02 mV at the lower punch. The force and distance were continuously monitored and documented by the sensors until the tablet was fully ejected from the die. The peak forces monitored from the force–displacement plots were designated as the compression and ejection forces. The thicknesses and diameters of the tablets were measured immediately after ejection using a digital micrometer (Mitutoyo 395-251, Mitutoyo, Tokyo, Japan). The occurrence of capping in the tablets was macroscopically examined by observing any capping phenomena during the friability and TBF tests.

#### 2.2.5. QC Test

##### Compaction Force Required to Fracture the MF-Loaded Tablets

The compaction force required to fracture the MF-loaded tablets was evaluated using a tablet combination tester (Multicheck VI; Erweka, Heusenstamm, Germany), which is widely used in hardness testing during pharmacopeial QC tests, as detailed in [10]. Although the term “hardness” is widely used and recognized in pharmacopeias, the more specific term “breaking force” was used in this study. The prepared tablets were positioned between two plates and compressed directly onto their surfaces, causing tablet fracture. The peak force monitored from the force–displacement plots was designated as the TBF.

##### Friability Test

Friability tests were conducted on the prepared tablets using a friability tester (PT F20E; Pharma Test, Hainburg, Germany), as detailed previously [10]. The tablets were then placed in a rotating drum and subjected to 100 rotations at 25 rpm. Weight loss was determined by weighing the tablets before and after the test.

### 2.3. Defect Prediction Method and System Configuration Using ML and DL

#### 2.3.1. ML and DL Model

RF is an ensemble ML algorithm based on decision trees that generates multiple decision trees and derives the final prediction by majority voting or averaging the predictions of each tree [15]. RF randomly selects only a subset of all input variables to find the optimal split criterion during splits at each node, thereby enhancing predictive performance by preventing overreliance on specific variables [16,17,18]. In this study, we aimed to achieve high classification accuracy and stability by capitalizing on the strengths of RFs and leveraging the multidimensional characteristics of the data.

An ANN is a supervised learning algorithm inspired by the neural network structure of the human brain and forms the foundational model upon which DL is built [19]. It mathematically models neurons and the connections between neural networks to learn from data and effectively evaluate the relationship between input and output data. An ANN typically consists of an Input Layer, one or more Hidden Layers, and an Output Layer [20]. ANN excels in processing non-linear data and effectively learns patterns within the data, thereby providing high predictive performance. In particular, it outperforms other techniques in dealing with multidimensional or complex relational data. In this study, utilizing these characteristics, a model was designed to learn the relationship between the input and output data, aiming to achieve high classification accuracy. The generalization performance of the model was optimized using the appropriate hidden-layer configurations and hyperparameter settings.

In this study, RF and ANN were chosen from various ML models. RF demonstrates robust performance even with non-linear data distributions and is useful for variable importance analysis, which offers excellent interpretability. Particularly as RF is less prone to overfitting compared with other ML models such as support vector machine (SVM) and Gaussian Process Regression (GPR), RF maintains stable performance even with small datasets. Therefore, in this study, RF was selected as a model suitable for analyzing important variables in tablet defect prediction during the manufacturing process. ANN excels in learning high-dimensional non-linear relationships and is more effective with large amounts of data. Although ANN is more difficult to interpret in terms of feature importance than RF, ANN has the advantage of learning more complex patterns using deep neural network structures. ANN, the smallest deep learning model, was selected as an initial approach to applying DL to the tablet defect detection system. Therefore, in this study, we compared and analyzed the performance of RF and ANN, evaluating the potential for tablet defect prediction by leveraging the strengths of each model and applying them to the TPM.

#### 2.3.2. Hyperparameter Optimization of RF and ANN Models

In this study, the Grid Search method was employed for hyperparameter optimization of the RF and ANN models. The optimized hyperparameters for each model are as follows: For RF, the number of trees (NumTrees) was optimized by testing values from 10 to 150 in increments of 10, and the maximum depth (MaxDepth) was varied from 2 to 10 in increments of 1, with the optimal values being NumTrees = 30 and MaxDepth = 4. For ANN, the learning rate (learningRates) was tested from 0.001 to 0.01 in increments of 0.001, the number of epochs (epochsList) ranged from 50 to 500 in increments of 50, batch sizes (batchSizes) varied from 2 to 128 in powers of 2, and dropout rates (dropoutRates) were tested from 0.2 to 0.5 in increments of 0.1. The optimal hyperparameters for ANN were found to be learningRates = 0.005, epochsList = 50, batchSizes = 32, and dropoutRates = 0.3.

#### 2.3.3. ROC-AUC Confidence Interval

To calculate the confidence interval for the ROC-AUC, the Bootstrap Resampling technique was used. This non-parametric method does not assume data normality and estimates the confidence interval through repetitive sampling. The number of repetitions was set to 2000, and the AUC values were used to determine the lower and upper limits of the confidence interval at the 2.5% and 97.5% percentiles, respectively. This method, known as the Bootstrap Percentile Method, was used for calculating a 95% confidence interval and is advantageous because it is not influenced by the data distribution, allowing for the estimation of a more generalizable confidence interval.

#### 2.3.4. Feature Importance Analysis

To assess the relative importance of each input variable, the permutation importance (PI) method was employed with out-of-bag (OOB) estimation. This technique involves individually shuffling the values of each feature in the trained model and measuring the corresponding increase in classification error, rather than RMSE, to determine the impact on the response variable [21]. To ensure the robustness of the results, the feature importance scores were averaged over 100 independent repetitions. By using OOB estimation, we avoided the need for a separate validation set, maintaining the model’s predictive ability without data leakage.

#### 2.3.5. Evaluation of the Results

A dataset containing 1482 data points was tested for TBF, with a standard deviation of 3.85 kp, and 1113 data points were tested for friability, with a standard deviation of 2.43%. The dataset was divided into training and test subsets, with 70% allocated for training and 30% reserved for evaluating the final predictive performance of the model on unseen data. In this study, 5-fold cross-validation (k = 5) was used to evaluate the model’s generalization performance, considering the small dataset size and learning stability [22]. Stratified random sampling was employed to ensure balanced representation of each class in both the training and test sets.

Based on the results, a confusion matrix was constructed and the performance of the model was evaluated [23]. The predictions were compared with the actual values and classified as True Positive (TP), False Positive (FP), True Negative (TN), or False Negative (FN). Using this classification, several evaluation metrics were considered, including Accuracy, Precision, Recall, F1-Score, and Fallout. Finally, ROC-Area Under the Curve (AUC) was employed as a threshold-independent performance evaluation metric, with an AUC value approaching 1 indicating superior classification performance [24].

Accuracy (Equation (1)) was employed to assess the overall performance of the model; however, in our dataset with class imbalance, it may not adequately reflect the true performance of the model. Therefore, other evaluation metrics were used in conjunction with these metrics.(1)$$Accuracy=\frac{TP+TN}{TP+TN+FP+FN}$$

Precision (Equation (2)) was used to reduce false defective predictions (FP) by considering the enhancement of the reliability of the products sorted as defective.(2)$$Precision=\frac{TP}{TP+FP}$$

Recall (Equation (3)) was employed to evaluate the sensitivity of defect detection and ensure that critical defects are identified during the manufacturing process.(3)$$Recall=\frac{TP}{TP+FN}$$

The F1-Score (Equation (4)), which is the harmonic mean of Precision and Recall, was used to assess the balance between these two metrics. This metric is more sensitive to lower values, ensuring that the errors (FP or FN) that occur in specific classes are not underestimated. Consequently, the model performance was fairly evaluated, even in our dataset with class imbalance.(4)$$F1-Score=2\times \frac{Precision\times Recall}{Precision+Recall}$$

Fallout (Equation (5)) was used as an auxiliary metric to measure the false-positive rate of the model and evaluate the accuracy and reliability of defect detection in abnormal data. A lower fallout value indicates a better capability of the model to accurately detect normal data, whereas a higher fallout value indicates a significant number of false positives, thus lowering the reliability of the model [25]. In pharmaceutical manufacturing, a higher fallout rate may lead to unnecessary QC costs and waste disposal. Conversely, a lower fallout indicates that the model is overly reliant on normal data, potentially causing it to overlook the defects.(5)$$Fall-out=\frac{FP}{FP+TN}$$

ROC-AUC is threshold-independent and was employed as a key metric for assessing the performance of the binary classification models.

To evaluate the sorting capability, the defective tablets sorted by the ML model program were visually inspected to verify whether they were accurately sorted as defective by the TPM. The TPM’s detection accuracy for the defects was calculated using the following equation:(6)$$Machine\;defect\;detection\;accuracy=\frac{N}{W}\times A$$
where *N* is the number of tablets correctly sorted as defective in the TPM compared with the sorting results of the programmed ML model, *W* is the total number of samples used for verification, and *A* is the defect-detection accuracy of the ML model.

#### 2.3.6. Configuration of the Defective Tablet Sorting Device

The configuration of the control system for in-line process sensing is shown in Figure 2. To apply the ML model to the TPM, the input variables were modified and mechanical parameters were simplified to include tablet compression speed, tablet ejection force, and compression force, which have many effects on tablet defects. The compression and ejection forces were measured using a compression load cell, and the rotational speed of the motor was measured to determine the compression speed. The data measured during compression were input into the control unit and stored in-line. Defective tablets were detected and sorted using a tablet-sorting device based on a trained ML model.

## 3. Results

### 3.1. Tablet Defect Prediction Using RF and ANN

#### 3.1.1. Data Preparation

For CQA prediction, the input variables used in the models included lubricant content, tablet weight, diameter, thickness, compression speed, compression force, and ejection force, as they may directly or indirectly affect tablet defects (Figure 3).

In the prediction, 1482 experimental data points were used to develop models for tablet defect prediction. The model performance was assessed using tablet defect prediction. The dataset was divided into 70% for training and 30% for testing to validate the model. Instead of min-max normalization, Z-score normalization was applied to standardize the data. To mitigate the influence of extreme values, the data were clipped within a range of ±3 standard deviations. In this study, although the ratio of non-defective to defective tablets was approximately 3:1, performance metrics such as Precision (91.92%), Recall (82.0%), F1-score (86.7%), and ROC-AUC (>0.89) for the RF model indicated minimal impact from data imbalance. Additionally, imbalance handling techniques were not applied, as the analysis of the Confusion Matrix showed balanced FN and FP rates across classes.

For the RF, the bagging technique was utilized to generate bootstrap samples through random sampling of the training data. Each decision tree was independently trained on these bootstrap datasets and individual predictions were performed. For the ANN, three fully connected layers were implemented in the hidden layer to learn complex patterns within the data. To introduce non-linearity, a ReLU (Rectified Linear Unit) activation function was used in each fully connected layer, whereas L2 regularization was implemented to enhance generalization and reduce the risk of overfitting [26]. The output layer utilized a sigmoid activation function to perform binary classification, and the final classification results were derived based on this output value.

#### 3.1.2. Criteria for Determining Tablet Defect

In this study, tablet defect criteria, which included two types of typical tablet defects related to tablet integrity—tablet capping occurrence during QC tests and inappropriate TBF—were determined. The classification of tablets as defective or non-defective was based on regulatory guidelines and prior experimental data.

Initially, the occurrence of tablet capping was evaluated macroscopically by observing capping during the TBF and friability tests. In addition, to determine the TBF criteria, experimental tests were conducted on the prepared tablets using a friability tester (PT F20E; Pharma Test, Hainburg, Germany) to ensure mechanical strength and friability (Figure 4). The tablets were placed in a rotating drum and subjected to 200 rotations at 25 rpm. Weight loss was determined by weighing the tablets before and after the test. Based on the experimental results, the lower limit of TBF was determined to be the TBF value corresponding to a friability value of 1.0% in the test, with reference to the USP guidelines [27]. Accordingly, the lower limits of the TBF for tablet defect detection models for Mg fractions of 0.6%, 1.3%, and 2.0% in the total mixture were determined to be 4.41 kp, 3.71 kp, and 1.74 kp, respectively. The upper limit of the TBF was determined to be 18 kp, based on comparisons with existing literature [28].

#### 3.1.3. Tablet Defect Prediction Results

Both the ANN and RF models demonstrated stable and excellent performance during the training and validation processes. In the ANN model, the training and validation losses decreased sharply during the initial epochs and then stabilized, with the training loss remaining slightly lower than the validation loss (Figure 5). This indicated that the model learned stably without overfitting (Figure 5 and Figure 6B). Similarly, the RF model maintained a consistent learning pattern according to the size of the training data set, demonstrating consistent accuracy without overfitting (Figure 6A).

The ROC curve analysis indicated that the AUC value for the RF model was 0.895, outperforming the AUC value of the ANN model (0.878), indicating that the RF model demonstrated an overall superior classification performance compared with the ANN model (Figure 7). Notably, the RF model exhibits a stronger capability to sort normal samples accurately, effectively reducing FN [29]. Although the ANN model showed a slightly lower AUC value than the RF model, its overall performance metrics were comparable, indicating stable and reliable classification performance.

The Confusion Matrix analysis demonstrated that both the ANN and RF models exhibited excellent classification performance (Figure 8). For the ANN model, the results were as follows: TP = 325, FN = 24, TN = 87, and FP = 9. In comparison, the RF model yielded TP = 326, FN = 20, TN = 91, and FP = 8. Based on these results, the calculated performance metrics were as follows: the ANN model achieved an accuracy of 0.9258, precision of 0.9061, recall of 0.784, F1-score of 0.841, and a fallout rate of 0.027. The RF model showed an accuracy of 0.9371, a precision of 0.9192, a recall of 0.820, an F1-score of 0.867, and a fallout rate of 0.024 (Table 3).

Considering the overall classification performance and practicality, the RF model was selected for application to the TPM for in-line defect detection in the manufacturing process. The RF model demonstrated better performance than the ANN model, achieving a higher AUC value and lower FN rate, thereby demonstrating robust accuracy in classifying both normal and defective tablets. In particular, the processing speeds of the RF model were faster than those of the ANN model, making it more suitable for in-line defect detection in TPM under manufacturing conditions. Based on these advantages, the RF model was selected as the applied model for the defect-detection system.

#### 3.1.4. Feature Importance Analysis of Factors Influencing Tablet Defects

The PI method was employed to evaluate the relative importance of each input variable in the predictive models. In this method, each input variable was individually shuffled, and the resulting change in classification error using OOB estimation was recorded. This approach allowed us to determine the influence of each variable based on its impact on model performance.

Figure 9 presents the relative importance results of each input variable for the RF model, as determined by predictions made on randomly selected data from the training set. The PI values were 0.598, 0.165, 1.990, 0.293, 0.316, 0.823, and 0.523, corresponding to MgSt fraction, tablet compression rate, compression force, ejection force, weight, thickness, and diameter, respectively.

According to the results obtained from the RF model, compression force was identified as the most influential individual factor affecting tablet defects among the input variables. Previous studies have described that applying higher compression forces during tablet formation reduces porosity and enhances interparticle bonding, resulting in tablets with greater mechanical strength, which aligns with our findings [30]. Additionally, high compression forces can trap air within the tablet matrix, and when air is not adequately released during compaction, this leads to internal stresses that cause capping upon ejection or during subsequent handling [31]. The MgSt fraction was ranked as the second most significant factor influencing tablet defects. As MgSt is hydrophobic, it tends to form a film over other excipients during mixing. This coating interferes with the bonding between particles, leading to reduced TBF and an increased risk of capping [32]. The impact of tablet thickness was also identified as higher than that of some other input variables. Variations in tablet thickness can influence porosity and density, leading to inconsistent mechanical properties and an increased risk of capping defects [12]. Based on the PI analysis, compression force, which was considered to have the most significant impact on the prediction of tablet defects, was included in the RF model as an input variable for the application of the TPM.

### 3.2. Verification of the In-Line TPM Application for the Detection of Defective Tablets

#### 3.2.1. Design and Manufacture of TPM for the Detection of Defective Tablets

A tableting system was developed to enable in-line defect sorting, as shown in Figure 10. The tablet compression process consisted of three stages: filling, compression, and ejection, during which data were collected and analyzed using an RF model program to determine whether a tablet was normal or defective (Figure 10A). The tablets identified as normal by the RF model program were directed to the normal discharge outlet, whereas defective tablets were sorted into a separate outlet, ensuring proper distinction from acceptable products (Figure 10B).

The main model machine was designed based on a commercial direct single-punch TPM with a customized mechanism design, structural analysis, and vibration analysis to determine the design variables before finalizing the detailed design. Subsequently, a sorting system was designed to sort defective tablets in-line using a dedicated guide chute during ejection (Figure 10C). Key components, including the main body and control units, were designed and manufactured, and a trial operation was initiated.

The drive system was designed as a gear train to enable precise control without slippage by incorporating reduced gearing. To reduce noise, the gear components meshed with the power source are made of Bakelite, a thermosetting synthetic resin, whereas the inner components are assembled using steel, which provides high torsional rigidity. This hybrid material structure effectively enhances the rigidity and noise reduction. For compression monitoring, a compression load cell (LCM375, FUTEK, Irvine, CA, USA) was installed in the middle of the main drive shaft, allowing in-line measurement of the compression and ejection forces during the tableting process. The tablet press speed was calculated by multiplying the number of tablets produced per unit time by the punch travel distance per cycle. In addition, a custom-designed sorting system was designed and manufactured to automatically detect and sort defective tablets, as shown in Figure 10C. The drive motor used was a 200 W stepper motor with fast response capabilities. To ensure smooth tablet movement, the base surface roughness of the die block was kept below 1.6 μm, and the discharge outlet was designed with a 15° inclined angle.

The final system specifications were confirmed as follows:

Maximum tablet compression force: 30 kN

Maximum tableting speed: 60 tablets/min

Tableting pressure tolerance: ±5%

Maximum filling depth: 15 mm

Maximum tablet diameter: 18 mm

#### 3.2.2. Verification of the TPM Application for the Sorting of Defective Tablets

The TPM was verified to detect and sort defective tablets in-line using the developed equipment. The tablets were compressed using MF-loaded granules, as described in Figure 1, with various ranges of tablet compression speed and pressure to randomly produce 700 types of tablets. The tablet compression speed, pressure, and ejection force were set between 10 and 75 mm/s, 2049 and 13,291 mPa, and 203 and 1974 mPa, respectively, to induce tablet defects (Figure 11).

Table 4 and Figure 12 present the verification results obtained from the sorting device based on the commands analyzed by the pretrained RF model program and sent to the machine. The defect-sorting accuracy was evaluated with 100 tablets per batch for seven trials; the machine-sorting accuracy was 100% for five trials, and two trials showed an error of one tablet each. The average machine-sorting accuracy was 99.43%. These results demonstrate that the developed model and equipment applied to the RF model program achieved a remarkably high performance. Based on these results, and considering the predictive accuracy of the pretrained RF model (93.71%) (Table 3), the accuracy of tablet defect detection based on the performance of the TPM is expected to be 93.17% for the manufacturing process batch. These results offer the possibility of detecting defects with over 90% accuracy using a non-destructive method in the manufacturing process, which will contribute to time savings and cost reductions in setting the initial phase of tableting for new drug development or scale-up research.

### 3.3. Practical Advantages and Applicability of the Proposed In-Line QC System

The proposed system offers a cost-effective and scalable alternative to conventional PAT tools by utilizing low-cost mechanical sensors, such as load cells and laser displacement sensors, instead of expensive spectroscopic equipment. Additional equipment required for running ML models in real time includes compression, distance, and rotation sensors, along with A/D amplifiers, a simple defective tablet sorting device, a 200-watt actuator, an industrial PC, and control software; however, the additional cost is expected to account for only a small proportion of the commercial tablet press used in our study. Once trained, the ML models can be applied in production lines without complex maintenance, and the RF and ANN models achieved high predictive accuracy (93.71% and 92.58%, respectively) with AUC values of 0.895 and 0.878. The integrated TPM demonstrated a defect detection accuracy of 93.17%, confirming its reliability for in-line QC. In terms of practical implementation, the system is easily integrated into standard manufacturing environments using existing process data, enabling real-time defect detection and sorting without interrupting production. The program and actuator processed and sorted the data in real time at the maximum speed of the developed TPM (one tablet/s) in this study without any trouble. Accordingly, the system is expected to be applicable to lab-scale production in the pharmaceutical industry, offering fast in-line processing through lightweight ML algorithms and low-power mechanical components, while reducing energy consumption and eliminating the need for separate off-line QC steps. Further investigations are required and planned to evaluate the system’s applicability to high-throughput rotary-type tablet presses for future large-scale production.

The performance and generalizability of the proposed ML models may vary with different APIs, excipients, tablet formations, and manufacturing conditions due to differences in the physicomechanical properties of the formulation, such as compressibility, plasticity, brittleness, and lubrication sensitivity. While the current models were trained on a specific formulation and MF of API, their adaptability can be enhanced by incorporating additional training data or applying techniques such as transfer learning. Since the system relies on universally applicable mechanical parameters such as compression force, compression speed, and ejection force, it is considered broadly usable across formulations with minimal hardware changes. Therefore, although some model retraining may be required, the system will be able to offer a scalable and robust approach for diverse tablet manufacturing environments.

## 4. Conclusions

In this study, we successfully developed and applied a TPM integrated with ML and DL-based models for in-line detection of defective tablets in pharmaceutical manufacturing. The RF and ANN models were trained using in-line process parameters to predict tablet defects, including capping and inappropriate TBF. The RF model demonstrated the highest predictive performance, achieving 93.71% accuracy with an AUC value of 0.895, followed by the ANN model with 92.58% accuracy and an AUC of 0.878. The newly designed TPM was applied to the trained RF model. The TPM successfully detected and sorted defective tablets in-line, achieving a sorting accuracy of 99.43% and a detection accuracy for defective tablets of 93.17%. These results validate the feasibility and effectiveness of the integrated system for non-destructive in-line defect detection. These findings highlight the potential of TPMs based on RF models as an effective, non-destructive PAT tool in pharmaceutical manufacturing. This approach enhances efficiency, reduces material waste, and ensures regulatory compliance. Future studies should optimize defect detection by applying various advanced models to the TPM and expanding the dataset to include various tablet formulations and process conditions.

## Figures and Tables

**Figure 1 pharmaceutics-17-00406-f001:**
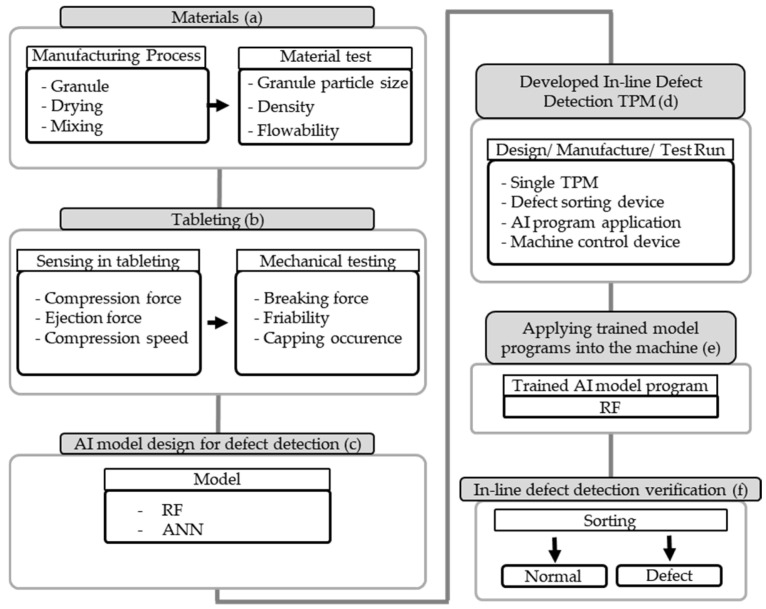
The procedure of design of defect prediction models using the RF and ANN and application to the TPM for in-line defect detection.

**Figure 2 pharmaceutics-17-00406-f002:**
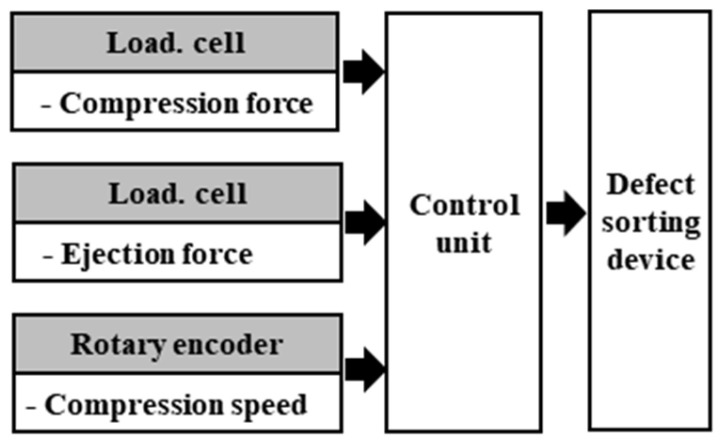
Configuration of In-line Data Input and Output.

**Figure 3 pharmaceutics-17-00406-f003:**
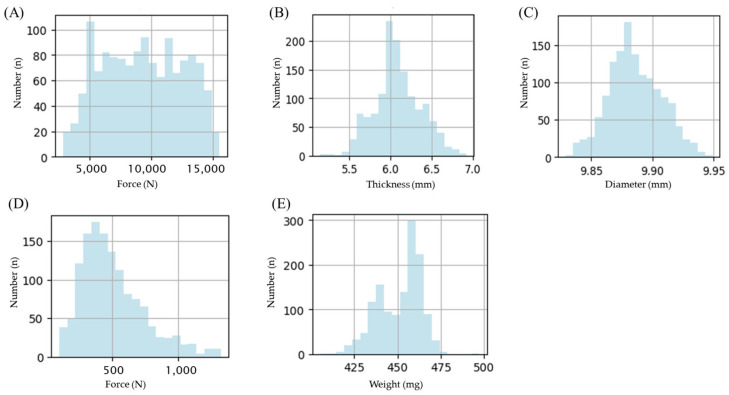
Histogram features of (**A**) compression force, (**B**) thickness, (**C**) diameter, (**D**) tablet ejection force, and (**E**) weight. Note: Data were obtained from 1482 types of tablets containing 1%, 2%, and 3% lubricant and compressed at speeds of 80 to 107 mm/s.

**Figure 4 pharmaceutics-17-00406-f004:**
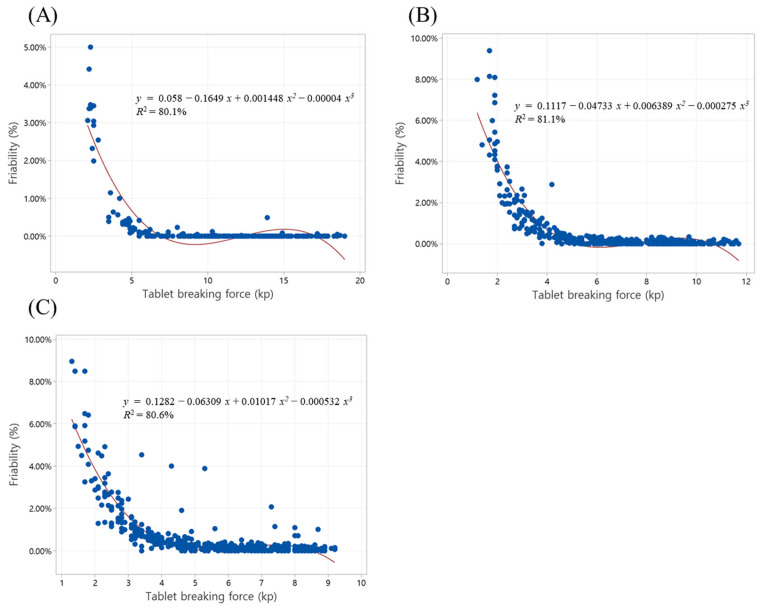
Correlation between TBF and the friability of tablets with Mg fractions of (**A**) 0.6%, (**B**) 1.3%, and (**C**) 2.0%. Note: Tests were performed on the tablets using a friability tester. The tablets were placed in a rotating drum and subjected to 200 rotations at 25 rpm. Weight loss was determined by weighing the tablets before and after the test. Note: The red line represents the fitted curve showing the relationship between TBF and friability.

**Figure 5 pharmaceutics-17-00406-f005:**
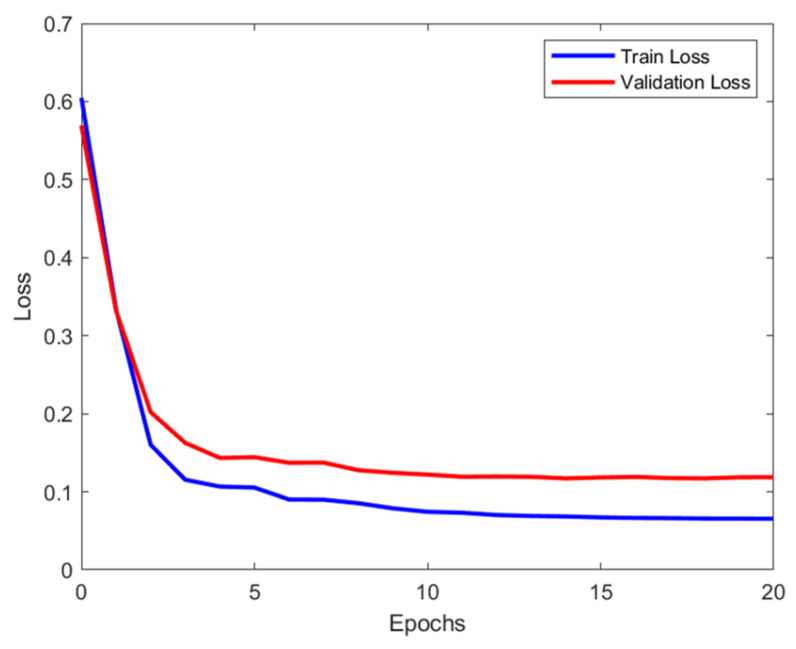
Loss graph for the ANN.

**Figure 6 pharmaceutics-17-00406-f006:**
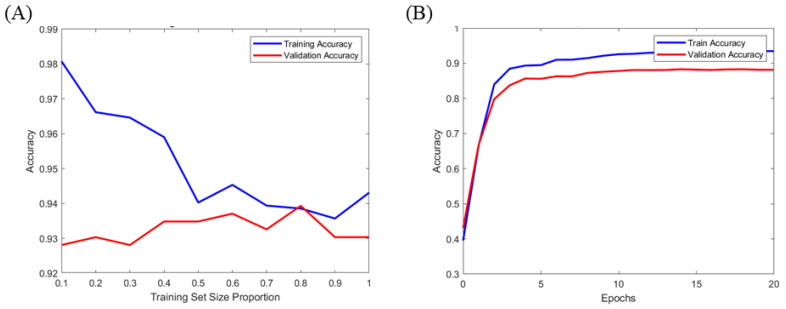
Accuracy graphs for the (**A**) RF model and (**B**) ANN model.

**Figure 7 pharmaceutics-17-00406-f007:**
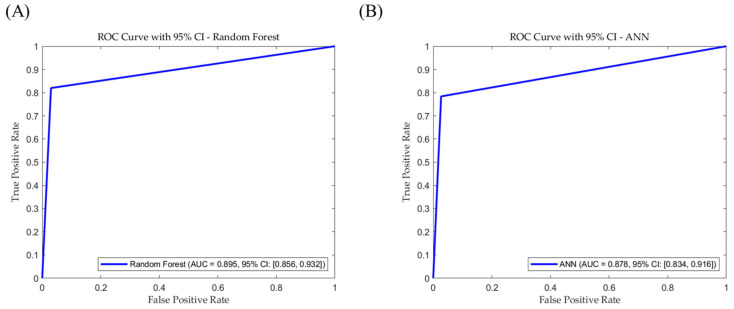
ROC and AUC graph for the (**A**) RF model and (**B**) ANN model.

**Figure 8 pharmaceutics-17-00406-f008:**
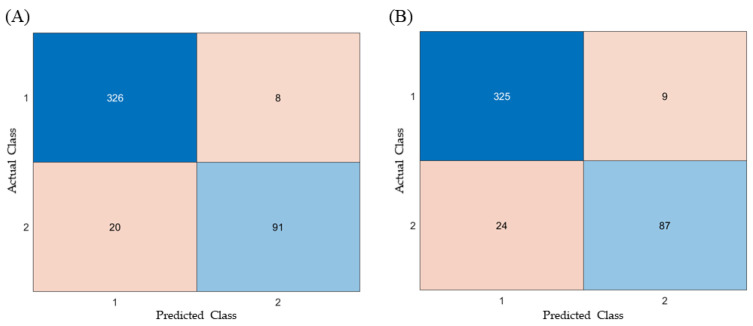
Confusion Matrix for test data in (**A**) RF model and (**B**) ANN model.

**Figure 9 pharmaceutics-17-00406-f009:**
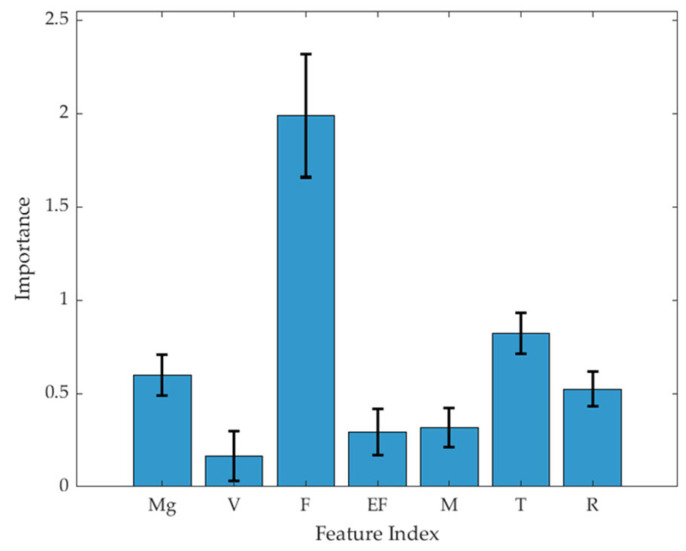
Permutation feature importance results for tablet defects based on the RF model. Data are represented as mean ± standard deviation (n = 100). Abbreviations: Mg, magnesium stearate; V, tablet compression speed; F, compression force; EF, ejection force; M, weight; T, thickness; R, diameter.

**Figure 10 pharmaceutics-17-00406-f010:**
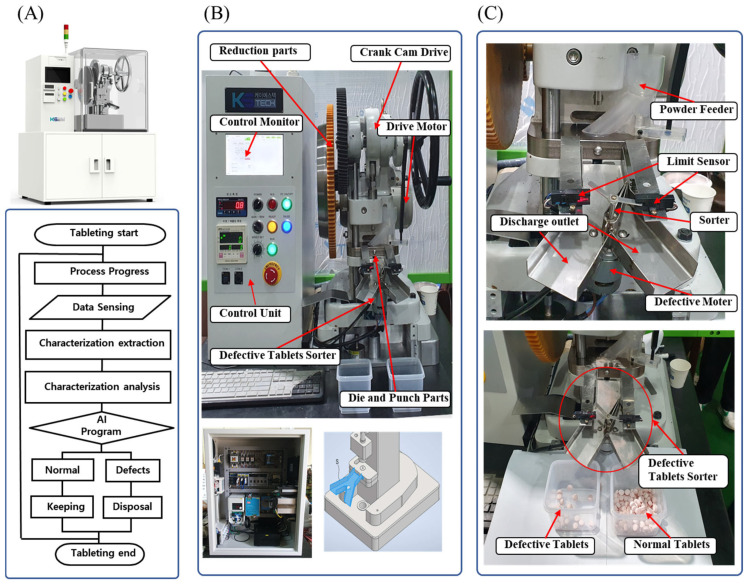
Custom-designed TPM for tablet defect detection: (**A**) TPM operation flow chart, (**B**) configuration of TPM, and (**C**) configuration of defective tablet sorting device.

**Figure 11 pharmaceutics-17-00406-f011:**
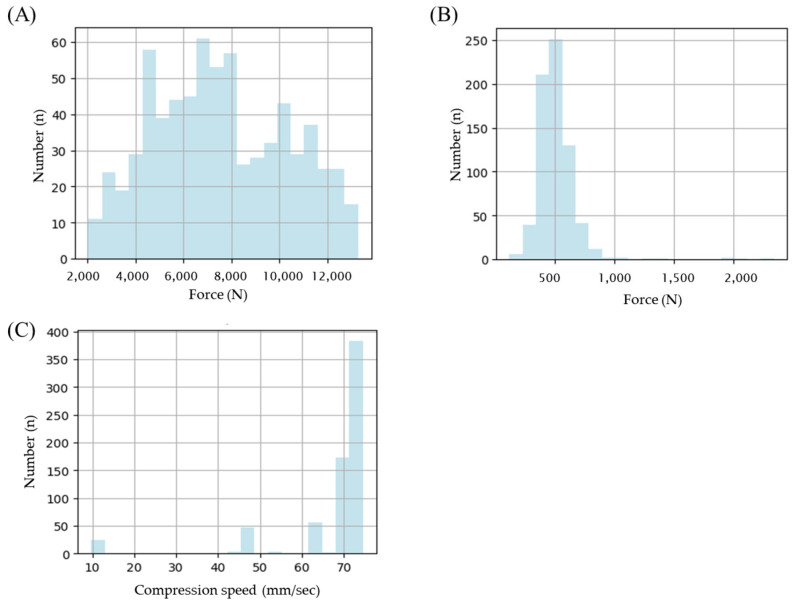
Histogram features of (**A**) compression force, (**B**) tablet ejection force, and (**C**) compression speed. Note: Data were obtained from 700 types of tablets containing 1% of lubricant.

**Figure 12 pharmaceutics-17-00406-f012:**
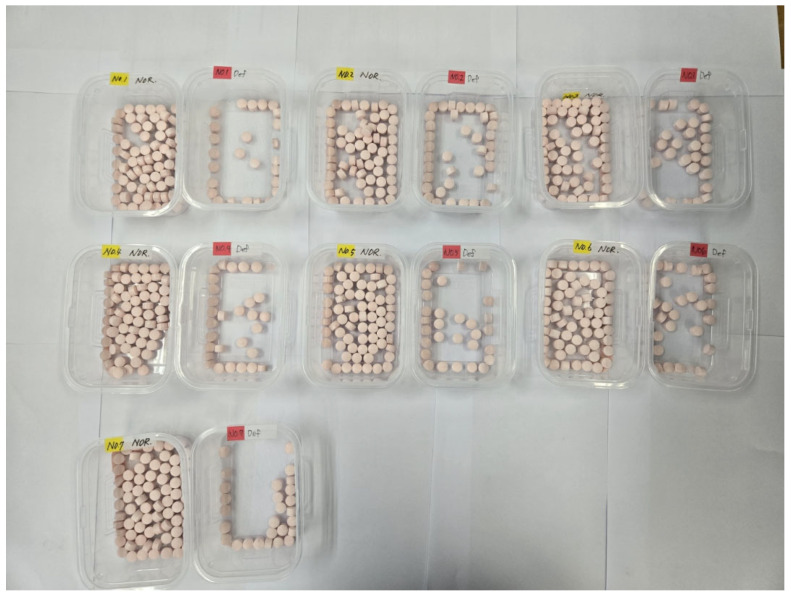
Images of normal and defective tablets sorted by the TPM based on the RF sorting model program. Note: The test tablets were colored pink due to the colorant (ferric oxide red). The tablets with yellow labels indicate normal tablets, whereas those with red labels indicate defective tablets.

**Table 1 pharmaceutics-17-00406-t001:** Physical characteristics of MF-loaded granules prepared via the wet granulation method.

Samples	MF-Loaded Granules
LOD (%) ^1^	0.74 ± 0.09
BD (g/mL) ^1^	0.43 ± 0.01
TD (g/mL) ^1^	0.48 ± 0.01
HR ^1,2^	1.12 ± 0.02
CI (%) ^1,3^	10.48 ± 1.90
Particle size d_0.1_ (μm) ^1,4^	10.73 ± 0.25
Particle size d_0.5_ (μm) ^1,5^	44.74 ± 2.19
Particle size d_0.9_ (μm) ^1,6^	110.93 ± 4.23
Span ^1,7^	2.24 ± 0.02

Abbreviations: BD, bulk density; CI, Carr’s index; HR, Hausner’s ratio; LOD, loss on drying; TD, tap density. ^1^ Data are presented as mean ± SD (*n* = 3); ^2^ HR was calculated by dividing the tapped density by the bulk density; ^3^ CI was calculated by dividing the difference between tapped density and bulk density by tapped density; ^4^ d_0.1_: volume-weighted diameter below which 10% of the total particles are found; ^5^ d_0.5_: volume-weighted diameter below which 50% of the total particles are found; ^6^ d_0.9_: volume-weighted diameter below which 90% of the total particles are found; ^7^ Span was calculated by dividing the difference between d_0.9_ and d_0.1_ by d_0.5_.

**Table 2 pharmaceutics-17-00406-t002:** Composition of the mixture used in the granulation process.

Function	Ingredient	Contents (mg)
Active pharmaceutical ingredient	MF	380
Binder	PVP K30	15.2
Lubricant	MgSt	2.74–9.35
Controlled release excipient	Carbomer 934P	7.6
Controlled release excipient	HPMC2208	38
Controlled release excipient	Methacrylic acid copolymer	15.2
Colorant	Ferric oxide red	0.3

Note: The composition of the mixture corresponds to a targeted tablet weight of 459 mg based on a MgSt content of 0.6%. The MgSt fraction was formulated in the range of 0.6% to 2.0% of the total mixture.

**Table 3 pharmaceutics-17-00406-t003:** Prediction accuracy for tablet defect for ANN models using input variables.

	RF	ANN
Accuracy	0.9371	0.9258
Precision	0.9192	0.9061
Recall	0.820	0.784
F1-Score	0.867	0.841
Fall-out	0.024	0.027

**Table 4 pharmaceutics-17-00406-t004:** Verification results of the RF model after the application of TPM.

No.	Number of Samples ^(a)^	RF Model Program		TPM		Machine-Sorting Accuracy (%) ^(d)^	Defect-Detection Accuracy for Model (%) ^(e)^	Machine Defect-Detection Accuracy (%) ^(f)^
		Normal	Defective ^(b)^	Normal	Defective ^(c)^			
1	100	75	25	74	26	98	93.71	91.83
2	100	62	38	62	38	100		93.70
3	100	64	36	62	38	98		91.83
4	100	71	29	71	29	100		93.70
5	100	70	30	70	30	100		93.70
6	100	63	37	63	37	100		93.70
7	100	74	26	74	26	100		93.70
Average	100	68.43	31.57	68	32	99.43		93.17

^a^ indicates the number of tablet samples for verification; ^b^ indicates number of defective tablet samples sorted by the RF model program; ^c^ indicates number of defective tablet samples sorted by the TPM; ^d^ indicates the accuracy of the TPM in correctly sorting defective tablets, as evaluated by comparing the sorting results with those of the programmed RF model; ^e^ indicates the detection accuracy of defective tablets for the RF model obtained from Table 3; ^f^ indicates the detection accuracy of defective tablets for the TPM.

## Data Availability

The data supporting the findings of this study are available from the corresponding author upon request.

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
