# Peer review of "Development of an Intelligent Tablet Press Machine for the In-Line Detection of Defective Tablets Using Machine Learning and Deep Learning Models"

_pharmaceutics, 2025, doi:10.3390/pharmaceutics17040406_

Round 1
Reviewer 1 Report
Comments and Suggestions for Authors
The manuscript entitled: Development of an Intelligent Tablet Press Machine for the In-Line Detection and Sorting of Defective Tablets Using Machine Learning and Deep Learning Models, presents an interesting and valuable study on the development of a tablet compression device equipped with machine learning and deep learning to detect the defective tablets during their manufacturing.
The topic is very interesting and I congratulate the authors for their work, as it is relevant to the field of pharmaceutical technology.
The manuscript is well written and structured, uses clear language and the results are well presented. The English language used is also very correct.
The methods for developing the smart tablet press are well described and provide a logical flow for the manuscript.
However, some points are not clear:
- What is the exact concentration of metformin in the final tablets? The expression used: The MF-loaded granules were formulated with an approximately 80% drug ratio to meet the high clinical dose (500–1,000 mg/day) is not clear. Considering that the final mass of the tablets is 450-475 mg (as shown in figure 3E), the dose is probably around 350 mg, which is not exactly a therapeutic concentration.
- The manufacturing method for tablets does not include a coating stage. Nevertheless, the authors state: Coating tablets need to have a certain degree of mechanical strength and friability to resist fracture during normal processing, handling, packaging… Please clarify.
- The authors mention the upper limit of the TBF, but the lower limit must also be stated
- Considering that the press machine is single-post eccentric, it is not clear why the compression speed is given in rpm.
- It is also not clear why the final tablets are pink, considering that metformin powder is white.
Reviewer 2 Report
Comments and Suggestions for Authors
The manuscript presents a novel approach to integrating machine learning (ML) and deep learning (DL) into the pharmaceutical manufacturing process for real-time defect detection and sorting of defective tablets. This study aligns well with current advancements in Process Analytical Technology (PAT) and Industry 4.0 initiatives in pharmaceutical production. The manuscript is well-structured and follows a logical progression. However, several key areas require clarification and improvement, particularly in the methodology, statistical analysis, and discussion of results.
- The study compares RF and ANN but does not provide sufficient justification for why these two models were chosen over other ML/DL models (e.g., Support Vector Machines, Gradient Boosting, or Convolutional Neural Networks for image-based defect detection).
- The authors should discuss whether the dataset was balanced in terms of defective vs. non-defective tablets. If the dataset was imbalanced, what techniques were used to mitigate this issue (e.g., Synthetic Minority Over-sampling Technique (SMOTE), class-weighting, or cost-sensitive learning)?
- For RF models, a feature importance analysis should be conducted and included to determine which factors (e.g., compression force, tablet thickness, ejection force) had the most significant impact on the prediction of tablet defects.
- The study mentions using k-fold cross-validation but does not specify the number of folds used.
- Did the authors test different hyperparameters, and if so, how were they optimized?
- The authors need to describe how the thresholds were determined for classifying tablets as defective or non-defective.
- Were these thresholds based on regulatory guidelines (e.g., USP, Ph. Eur.) or prior experimental data?
- The authors need to mention how many tablets were tested for TBF and friability. What was the standard deviation of these measurements?
- The authors need to mention if the hardness test was performed using a universal testing machine or another instrument.
- How do the results compare with existing literature on defect thresholds in tablet manufacturing?
- The authors need to mention how this proposed system compares with existing PAT technologies like Terahertz imaging, Raman spectroscopy, or Near-Infrared Spectroscopy (NIR)?
- The authors need to discuss a comparison of cost, accuracy, and practical implementation.
- Can the ML models generalize to different tablet formulations, excipients, or manufacturing conditions?
- How would the system perform with different active pharmaceutical ingredients (APIs) with varying compression properties?
- The study does not discuss the computational cost of running ML models in real time.
- How feasible is this system for large-scale production in terms of processing speed and energy consumption?
- There are minor typographical errors throughout the manuscript. Example: *"Pharmaceutics 2025, 17, x FOR PEER REVIEW 2 of 17"* appears incorrectly.
- Figures 6 and 7 (ROC-AUC curves) should include confidence intervals for better statistical reliability.
Round 2
Reviewer 1 Report
Comments and Suggestions for Authors
Thank you for your replies
Reviewer 2 Report
Comments and Suggestions for Authors
The authors responded to all comments appropriately.